# Electrospun Silk Fibroin/Polylactic-co-glycolic Acid/Black Phosphorus Nanosheets Nanofibrous Membrane with Photothermal Therapy Potential for Cancer

**DOI:** 10.3390/molecules27144563

**Published:** 2022-07-18

**Authors:** Xia Li, Jiale Zhou, Haiyan Wu, Fangyin Dai, Jiashen Li, Zhi Li

**Affiliations:** 1State Key Laboratory of Silkworm Genome Biology, Southwest University, Chongqing 400715, China; lx132465@email.swu.edu.cn (X.L.); zzz18771922@163.com (J.Z.); why19823327060@163.com (H.W.); fydai@swu.edu.cn (F.D.); 2Chongqing Engineering Research Center of Biomaterial Fiber and Modern Textile, College of Sericulture, Textile and Biomass Science, Southwest University, Chongqing 400715, China; 3Key Laboratory for Sericulture Biology and Genetic Breeding, Ministry of Agriculture and Rural Affairs, Southwest University, Chongqing 400715, China; 4Department of Materials, The University of Manchester, Manchester M13 9PL, UK

**Keywords:** black phosphorus nanosheets, silk fibroin, cancer, photothermal therapy, electrospinning

## Abstract

Photothermal therapy is a promising treating method for cancers since it is safe and easily controllable. Black phosphorus (BP) nanosheets have drawn tremendous attention as a novel biodegradable thermotherapy material, owing to their excellent biocompatibility and photothermal properties. In this study, silk fibroin (SF) was used to exfoliate BP with long-term stability and good solution-processability. Then, the prepared BP@SF was introduced into fibrous membranes by electrospinning, together with SF and polylactic-co-glycolic acid (PLGA). The SF/PLGA/BP@SF membranes had relatively smooth and even fibers and the maximum stress was 2.92 MPa. Most importantly, the SF/PLGA/BP@SF membranes exhibited excellent photothermal properties, which could be controlled by the BP@SF content and near infrared (NIR) light power. The temperature of SF/PLGA/BP@SF composite membrane was increased by 15.26 °C under NIR (808 nm, 2.5 W/cm^2^) irradiation for 10 min. The photothermal property of SF/PLGA/BP@SF membranes significantly killed the HepG2 cancer cells in vitro, indicating its good potential for application in local treatment of cancer.

## 1. Introduction

Cancer is a serious disease characterized by uncontrolled growth and fast proliferation of abnormal cells. Cancer cells have ability to spread to different tissues and organs in the human body, finally causing death. Recent statistics reveal that cancer is the first- or second-ranked fatal disease for people over 70 years old, in 112 countries out of 183 countries [1,2]. Despite the tremendous efforts that are devoted to its treatment, cancer remains an unsolved problem in clinical therapy. So far, there are four treatment methods for cancers, including chemotherapy, radiotherapy, immunotherapy, and phototherapy [3,4]. Chemotherapy and radiotherapy are still commonly used alongside surgery to treat cancers. However, the random distribution of the drug molecules in the organism can inflict a great threat to normal tissues and organs. Meanwhile, the patients have to take several consecutive doses of the drugs to achieve the desired therapeutic effect, which greatly increases the financial burden on patients. Immunotherapy is safe, effective and, at the same time, makes it impossible for the cancer cells to escape from the immune recognition mechanism, which eventually eliminates the cancer cells. However, the immune cells are unable to kill tumor cells in many cases, even if the tumor cell-specific antigens have been recognized by the T cells [5,6,7]. Photothermal therapy (PTT) is a combination of the local administration of photothermal agents and near-infrared laser irradiation (NIR), in which a nanoscale transducer converts photonic energy into heat, and then this locally released heat effectively kills the cancer cells. With high efficacy and low side effects, this treatment has attracted extensive attention in the latest research works [8].

Many nanoparticles, such as gold nanoparticles, manganese oxide nanoparticles, and selenium-coated tellurium nanoparticles, have been used as photothermal agents to achieve the maximal ablation of cancer cells [9,10]. However, their poor degradability may lead to long-term cytotoxicity in the organism before being metabolically removed. Therefore, biodegradable photothermal agents are urgently needed for clinical use. Since black phosphorus (BP) nanosheets were first discovered in 2014, they have become one of the most attractive nanomaterials. BP nanosheets have many unique properties, such as tunable band gap, excellent surface activity, extensive spectrum light absorption, and excellent photothermal conversion efficiency. Most importantly, the BP nanosheets have excellent biocompatibility and biodegradability, and their degradation products are non-toxic phosphates and phosphonates [11,12]. Therefore, BP nanosheets can be used as a photothermal agent for cancer treatment. Shao et al. prepared biodegradable nanospheres of black phosphorus quantum dots (BPQDs) and polylactic-co-glycolic acid (PLGA) for tumor treatment. After intravenous injection into the body, the BPQDs/PLGA nanospheres reduced the degradation of the BPQDs during prolonged cycling to ensure sufficient MCF7 breast tumor accumulation for efficient PTT [13]. Li et al. presented a combination of BP nanosheets and gemcitabine (GEM) to prepare a thermosensitive hydrogel for chemo–photothermal combination therapy against cancer. The hydrogel effectively killed the 4T1 tumors in vivo. Meanwhile, the BP-GEM-GEL showed negligible systemic toxicity [14]. The BP nanosheets are expected to be the next generation of NIR light-mediated nanophotonic materials for the photothermal or photodynamic therapy of tumors [15,16].

Although the BP nanosheets have many excellent properties, there are still some technical obstacles in preparing BP nanosheets [17]. N-methyl pyrrolidone was generally used as an exfoliating agent for the liquid stripping of the BP nanosheets, which is not eco-friendly. The BP nanosheets produced by this method are prone to be degraded by oxygen and water in the physiological environment, resulting in negative effects on their optical and electronic properties [18,19]. Furthermore, the poor solution-processability of BP nanosheets limits the fabrication of multi-structured, functional BP-based materials. An efficient and environmentally friendly method has been developed to synthesize thin-layered BP nanosheets in aqueous media through a strong binding between silk fibroin (SF) and 2D nanomaterials. As an exfoliating agent, SF provides a long-term dispersion stability of the resulting nanosheets in a physiological environment [20]. The prepared SF-modified BP nanosheets (BP@SF) show long term stability and good solubility, so that it has the potential for good applications in the biomaterial filed.

It is still a challenge to effectively deliver BP nanosheets to cancerous areas, since the conventional systemic administration of free BP nanosheets by intravenous injection is associated with a high propensity for deposition in normal tissues [21]. Therefore, to enhance the cancer specificity and mitigate the adverse effects on healthy organs, immobilizing the BP nanosheets in scaffolds for local treatment may enhance the clinical meaningfulness [22,23,24]. Electrospinning is a technology to produce fibrous and porous polymer scaffolds. The electrospun materials have a large field of potential application, owing to their large surface area, controllable surface configuration, well-modified surface, complex pore structure and good biocompatibility [25,26,27,28,29,30]. Electrospun membranes are widely used in the biomedical field, including tissue engineering, drug delivery, and wound healing. Chen et al. doped polyaniline nanoparticles into poly(ε-caprolactone) and gelatin (PG) to form nanofibrous fabrics. The polyaniline PG was implanted directly onto the surface of hepatoma H22 tumors and excellent anti-tumor effects were observed [31]. Wang et al. designed a tissue-engineered membrane by incorporating Cu_2_S nanoflowers into biopolymer fibers, based on a modified electrospinning method. With uniformly embedded Cu_2_S nanoparticles, the membrane resulted in over 90% mortality of skin tumor cells under NIR irradiation and effectively inhibited tumor growth in mice [28]. These recently reported studies prove that electrospinning technology is good for loading materials for photothermal therapy.

Inspired by Huang et al. [20], this study prepared thin-layered BP nanosheets, using an SF-assisted liquid exfoliation method. The SF-modified BP nanosheets (BP@SF) maintained excellent photothermal properties. Then, the BP@SF was introduced into polymer fibers by electrospinning. Three composite nanofiber membranes were prepared with PLGA, SF, and different contents of BP@SF. The mechanical properties and biocompatibility of SF/PLGA/BP@SF were investigated. More importantly, the photothermal therapy of SF/PLGA/BP@SF on tumor cells was examined in the near infrared (808 nm, 2.5 W/cm^2^).

## 2. Results and Discussions

### 2.1. Preparation and Characterization of BP@SF Sheets

The BP@SF sheets were prepared by ultrasound-assisted liquid exfoliation, using SF as an effective exfoliator. With a unique hydrophilic–hydrophobic structure and abundant carboxyl groups, SF can facilitate the interfacial bonding with 2D nanomaterials [32,33]. On this basis, during the exfoliation process, the SF molecules stably bound to the BP crystal surface through strong hydrophobic interactions. Meanwhile, their hydrophilic regions are exposed to water to stabilize the exfoliated BP nanosheets and prevent re-agglomeration. The ultrathin BP@SF sheets were obtained after sonication of BP powders in aqueous SF solution for 2 h. The size of BP@SF was from 200 to 400 nm (PDI = 0.688) (Figure 1a). The SEM observation (Figure 1b) and EDS analysis (Figure 1c) showed that the BP@SF were well distributed. The exfoliated BP@SF exhibited a lamellar appearance and the lattice spacing of the sheets was 0.2823 nm (Figure 1d,e), which was consistent with the study reported [20].

FTIR was used to analyze the chemical structure of BP@SF (Figure 2a). For the SF powders, the peaks at 1640 cm^−1^, 1405 cm^−1^, 1274 cm^−1^, and 631 cm^−1^ belong to amide I, amide II, amide III, and amide V, respectively [34]. Meanwhile, the BP@SF curve exhibits peaks at 1640 cm^−1^ (amide I of SF), 1405 cm^−1^ (amide II of SF), 1274 cm^−1^ (amide III), and 631 cm^−1^ (amide V), demonstrating that the SF molecule is stably bound to the surface of the BP crystal. The XRD pattern of BP@SF shows the presence of the characteristic peak of BP (Figure 2b), which matches its orthorhombic structure (Figure 2c) [35]. This further confirms the successful binding of BP to the SF.

BP has a wide spectrum of light-absorption qualities. It absorbs light between 400 and 900 nm. As a result, BP is frequently utilized as a photothermal agent in the PTT of cancer. Near-infrared light at 808 nm is commonly utilized to test the photothermal characteristics of BP [36,37]. In this study, we investigated the spectrum absorption of 0.1 mg/mL, 0.2 mg/mL, and 0.4 mg/mL BP@SF in an aqueous solution between 400–900 nm, with an emphasis on the light absorption between 775 and 855 nm (Figure 2(e_2_, e_3_)). BP@SF had a broad range of light absorption between 400 and 900 nm (Figure 2(e_1_)), with the absorption peak from 800 to 810 nm being the most prominent (Figure 2e). Furthermore, as the concentration of BP@SF grows, the intensity of the light absorption increases continually. As a result, NIR at 808 nm was used for further testing in this study.

PTT is a relatively safe method of treating cancer by inducing apoptosis through local heating, with minimal negative effects on the surrounding healthy tissue [8]. BP nanosheets are a favorable PPT agent. To demonstrate the NIR photothermal properties of BP@SF, the aqueous solution without BP@SF (control), 0.1% BP@SF aqueous solution, and 0.2% BP@SF aqueous solution are irradiated with a NIR (808 nm, 0.5 W/cm^2^) for 10 min and then the heating temperature is measured. The results show that the photothermal effect of the control group is weak, with an increase of only 2.5 °C (Figure 2d). The heating effect increases significantly with the increased BP@SF concentration. In the aqueous solution containing only 0.1% BP@SF, the heating effect is ten times greater than that of the control. More strikingly, a heating effect of 37 °C could be achieved in an aqueous solution containing 0.2% BP@SF. This suggests that BP@SF can effectively convert light energy into heat energy, and has the potential to be used as an agent in PTT to ablate cancer cells.

### 2.2. Preparation and Characterization of BP@SF Electrospun Membranes

Based on the good solution processability of BP@SF, four fibrous membranes were electrospun with 0 mg BP@SF, 1 mg BP@SF, 2 mg BP@SF, and 4 mg BP@SF and named SF/PLGA, SF/PLGA/BP@SF-1, SF/PLGA/BP@SF-2, and SF/PLGA/BP@SF-4, respectively. There were 0.6 g of SF and 0.6 g of PLGA in each membrane. The SEM observation (Figure 3a) showed that the electrospun SF/PLGA fibers have a smooth appearance and the average fiber diameter was approximately 470.40 ± 96.17 nm, while the electrospun fibers with loaded BP@SF had a relatively rough surface and their average fiber diameter varied (Figure 3b). The fiber diameter of SF/PLGA/BP@SF-1 ranged from 200 nm to 800 nm. With a further increase in the loaded BP@SF content, the fibers ranged from 300 nm to 900 nm and the average diameter of the fibers significantly increased, from 444.60 ± 90.20 nm (SF/PLGA/BP@SF-1) to 607.60 ± 116.93 nm (SF/PLGA/BP@SF-4) (Figure 3b). Thus, the mixing of BP@SF greatly affects the morphology of the electrospun membrane. This is because the viscosity of the electrospun-blend solution increases with the increasing BP@SF concentration. In general, the viscosity of the electrospinning solution affects the nanofibrous morphology and fiber diameter, and the viscosity of the solution can be adjusted by the polymer concentration, together with the blended material [38,39]. The higher viscosity of the electrospinning solutions with higher BP@SF content leads to a significant increase in the diameter of the electrospun fibers [40,41].

The BP@SF elements on the electrospun membrane were characterized using elemental analysis techniques (Figure 3c). The EDS analysis indicated that the content of the P element was 0.13% in SF/PLGA (Table 1), which was from the SF protein. With the loaded BP@SF, the P element content increased and reached up to 0.25% for SF/PLGA/BP@SF-4 (Figure 3c and Table 2), which identified that the BP@SF was evenly loaded on the fibrous membrane through electrospinning.

### 2.3. Hydrophilic and Hydrophobic Properties

Hydrophobicity is one of the important properties of electrospun fibrous membrane [42]. The hydrophilic properties of the four composite electrospun membranes are investigated by the contact angle tests (Figure 4a). With increasing time, the contact angles of all of the four membranes gradually decrease from >100° to 0°. A total of 12 s is required for the SF/PLGA membrane to have a 0°. For the SF/PLGA/BP@SF-1 (SF/PLGA/BP@SF-2) and SF/PLGA/BP@SF-4, the time is 20 s and 30 s, respectively. These results indicate that all of the electrospun membranes have good hydrophilicity. Furthermore, the contact angle of SF/PLGA is the minimum at each moment, the blended BP@SF makes the contact angle of the electrospun membrane become bigger at the same time point. Most notably, at the third second, the contact angles of SF/PLGA, SF/PLGA/BP@SF-1, SF/PLGA/BP@SF-2, and SF/PLGA/BP@SF-4 are 28.53 ± 3.75°, 75.40 ± 3.49°, 118.37 ± 2.74°, and 120.37 ± 4.94°, respectively. The higher the content of the BP@SF, the bigger the contact angle of the electrospun fibrous membrane. Although the BP@SF has good stability in water, the surface of the BP nanosheets is hydrophobic, according to isotope experiments, contact angle measurements, and calculations [43]. Therefore, as the concentration of the BP@SF increases, the contact angle gradually increases, and the hydrophilic properties of the electrospun membrane become weaker.

### 2.4. Tensile Properties

Appropriate mechanical properties of electrospun membrane are necessary for its application [44,45]. Therefore, the effect of the BP@SF on the mechanical properties of the electrospun membrane were investigated, using a tensile testing machine (Figure 4c). Among these electrospun membranes, the SF/PLGA membrane had the lowest stress, strain, and Young’s modulus, which were 1.51 ± 0.89 MPa, 229 ± 3.3% MPa, and 254.83 ± 3.37 MPa, respectively. Compared to the SF/PLGA electrospun membrane, the breaking strength and elongation at break of the electrospun membrane increased correspondingly with the increase in the BP@SF content in the electrospun membrane. In particular, the stress, strain, and Young’s modulus of the SF/PLGA/BP@SF-4 membrane enhanced to 2.92 ± 0.32 MPa, 51 ± 2.3% and 104.94 ± 2.15 MPa, respectively. This may attribute to the good mechanical properties of the BP nanosheets. It was proved that the BP had good mechanical properties and can enhance the mechanical properties of the composites [46].

### 2.5. Photothermal Effects and Stability Tests

For PTT, photothermal characteristics are essential [26]. The electrospun membranes were irradiated with different powers of NIR in the dry (air) state, and the temperature changes and thermal images were recorded (Figure 4b). The electrospun membranes with different concentrations of BP@SF show a temperature increase under NIR irradiation at 0, 0.5, 1, 1.5, 2, and 2.5 W for 10 min. All of the three electrospun membranes show the highest temperature rise at 2.5 W NIR irradiation. Among them, the temperature of the SF/PLGA/BP@SF-4 membrane increases most, with a temperature increase of 15.26 °C from the room temperature at 2.5 W NIR power. The SF/PLGA/BP@SF-1 membrane rises by 8.34 °C and the SF/PLGA/BP@SF-2 membrane rises by 12.89 °C under the same conditions. These results indicate that the SF/PLGA/BP@SF-4 membrane has great and controllable photothermal properties and that the heating effect increases with the increasing BP@SF concentration. Thus, adjusting the BP@SF concentration and the NIR power to control the temperature can realize the PTT, so that they have great potential for application in cancer therapy.

The stability of NIR absorption in a wet environment is closely related to the effectiveness for tumor therapy. The stability of SF/PLGA/BP@SF-4 to NIR absorption properties in the PBS solution was tested. The temperature change of the solution was recorded under five on/off irradiations of the NIR light (808 nm, 2.5 W/cm^2^) (Figure 4d). The temperature of the solution increases by 15.5, 14.6, 13.9, 13.5, and 13.1 °C after five NIR irradiations for 10 min. Although the enhanced temperature slightly decreases, the SF/PLGA/BP@SF-4 also exhibits a stable photothermal property after cycle numbers of test.

### 2.6. Biocompatibility

The mouse fibroblast cell line L929 was used to assess the biocompatibility of the electrospun membranes. The cells were cultured on each electrospun membrane for 3 days and the cell activity and cytotoxicity were examined by CCK-8 assay and double fluorescent staining with Calcein-AM /PI (Figure 5a). In general, the cells have normal growth and proliferation during the culture days. With the increase in inoculation time, the cell OD values of all of the groups show a rising trend. However, compared to the control group, there is a slight decrease in the cellular OD values for the SF/PLGA/BP@SF membranes on the third day. This suggests that the addition of BP@SF may have a slight inhibitory effect on the cell proliferation. Furthermore, the LIVE/DEAD cell-staining images show that none of the three SF/PLGA/BP@SF membranes causes cell death, similar to the results of the control group (Figure 5b). This indicates that all of the membranes are not toxic and have good biocompatibility.

### 2.7. In Vitro Anti-Tumor Properties

Human hepatocarcinoma cell line (HepG2) was inoculated onto the membranes and the anti-tumor effect of the SF/PLGA/BP@SF membranes under the photothermal effect was investigated in vitro (Figure 6a). On the first day, the cell viability showed no significant difference between the five groups without NIR treatment. After being treated by NIR (808 nm, 2.5 W/cm^2^ for 10 min), the cell viability dramatically decreases for SF/PLGA/BP@SF-1-L, SF/PLGA/BP@SF-2-L, and SF/PLGA/BP@SF-4-L, respectively, while the cell viability of SF/PLGA-L group had no significant reduction. It was evident that the BP@SF on fibrous membrane converts the NIR to heat, which inhibits the cancer cell growth and proliferation, leading to the lower cell viability. There was a similar finding for cell viability on the second day. Further, the LIVE/DEAD cell staining was conducted to observe the cell stadium after treating by NIR (Figure 6b). The cells of the control and SF/PLGA groups show green fluorescence and almost all of the cells are alive, while a great numbers of dead cells for the SF/PLGA/BP@SF-1-L group, SF/PLGA/BP@SF-2-L group, and SF/PLGA/BP@SF-4-L group are found (shown in red), indicating that the photothermal effect of these groups can significantly kill the HeGp2 cancer cells. These results demonstrate that the SF/PLGA/BP@SF membranes have excellent and controllable photothermal properties, which have a good potential for application in cancer therapy.

## 3. Discussion

With the development of medical technology and the demand for precision in treatment, conventional cancer treatments, such as chemotherapy, face the fundamental problems of strong side effects for the normal cells and a constant recurrence, which seriously affect the long-term effects of cancer treatment. Precise and effective treatments remain a challenge and also an ultimate goal of cancer treatment. PTT, which relies on photothermal properties for converting light irradiation into heat to ablate tumors, has attracted wide attention in recent research in cancer treatment, owing to its non-invasive, local treatment and good therapeutic effects. The preclinical studies and early trials prove that PTT has good therapeutic effects on superficial tumors (e.g., skin cancer, head and neck cancer) [47].

In general, the therapeutic effect of PTT mainly depends on photothermal materials. Under the local irradiation of NIR at a certain wavelength, photothermal materials can quickly switch from ground state to excited state and then dissipate back to ground state in the form of heat. The heat dissipated during this process acts on the cancer cells, leading to cell apoptosis and cell death. The NIR photothermal materials have the advantage of deep penetration in the body, enabling less biological interference and better treatment in deep tissues. In addition, the biological cells and tissues have little absorption in the NIR wavelength region, and thus are not damaged under low-intensity NIR laser [10]. Depending on clinical needs, ideal photothermal materials need to have excellent biodegradability, good photostability, broad near-infrared spectral absorption, and high photothermal conversion rates [9,48]. Compared with other photothermal agents [49], BP nanosheets have the characteristics of non-toxic biodegradability, excellent biocompatibility, and a high photothermal conversion rate [50], which make them have a promising clinical application [51].

In this paper, BP nanosheets were exfoliated with SF auxiliary liquid by means of a green and environmentally friendly liquid-phase exfoliation method [20]. Then, the BP@SF was mixed with PLGA and SF to prepare electrospun fibrous membranes by electrospinning. With the addition of the BP@SF, the viscosity of the electrospinning solution increased, which in turn increased the diameter of the electrospun fibers, affecting the pore size of the electrospun membrane and further affecting cell adhesion and distribution. In addition, owing to the excellent mechanical properties of BP@SF, the mechanical properties of the electrospun membrane were enhanced with the addition of BP@SF. Three different sizes of BP nanosheets (large BP (394 ± 75 nm), medium BP (118 ± 22 nm) and small BP (4.5 ± 0.6 nm) were prepared by Fu et al. using a modified liquid stripping technique [52]. The larger BP nanosheets had the best thermal ablation effect on cancer cells. At a concentration of 2.5%, the temperature of the larger BP solution could be increased by 24.0 °C after 10 min of irradiation with the 808 nm laser. In comparison, the temperature rise of the medium BP and small BP solutions was 21.8 °C and 19.2 °C, respectively. For the preparation method in this study, the size of the BP@SF was relatively large for these three sizes, and it was between 400 and 600 nm. Only 0.2% of the BP@SF concentration can increase the temperature to 37 °C in an aqueous solution, exhibiting excellent photothermal conversion properties. Meanwhile, the SF/PLGA/BP@SF-4 electrospun membrane also demonstrated excellent photothermal conversion performance. Under NIR (808 nm, 2.5 W/cm^2^) irradiation, the temperature of the electrospun membrane increased by 15.26 °C. In addition, under NIR (808 nm, 2.5 W/cm^2^) irradiation, the cell viability of the SF/PLGA/BP@SF-1-L decreased sharply, and the OD value was significantly lower than that of the other groups under the same conditions. These results indicate that SF/PLGA/BP@SF-1-L has a significant inhibitory effect on HepG2 cells under photothermal properties. Further, the addition of BP@SF did not affect the biocompatibility of the membrane. Phosphorus, one of the most common biological components of the human body, is closely associated with the formation of cell membranes and deoxyribonucleic acid (DNA), as well as minerals. The lone pair of electrons on each phosphorus atom results in a high reactivity of the BP nanosheets to water and oxygen, leading to the degradation of the BP nanosheets to non-toxic phosphates (PO_4_^3−^) and other P_x_O_y_ in the physiological environment [53]. The PO_4_^3−^ captures surrounding positive calcium ions (Ca^2+^) to form calcium phosphate (CaP) [54], which promotes local biomineralization for in situ bone regeneration [55].

In conclusion, the SF/PLGA/BP@SFs electrospun membranes can be simply fabricated with excellent mechanical properties and good biocompatibility, controllable photothermal conversion properties under NIR, and effective ablation of cancer cells, which gives them great potential in the treatment of cancer (Figure 7). However, there are also some challenges in this study. For example, only in vitro cellular experiments and cancer cell ablation experiments were performed. In vivo animal studies are required for further evaluation.

## 4. Materials and Methods

### 4.1. Materials

The Bombyx mori silkworm cocoons were obtained from Southwest University. The (BP) (purity > 99.0%) was obtained from Feynman Nanomaterials Technology Co. The polylactide-co-glycolide (PLGA, 50/50, MW: 110,000) was acquired from Jinan Daigang Biotechnology Co., Ltd. (Jinan, China). The chloroacetic acid (CA) (purity = 98.0%) and potassium bromide (KBr) were obtained at McLean (Shanghai, China). The phosphate buffer solution (PBS) was obtained from Thermo Fisher (Shanghai, China). The sodium carbonate anhydrous and anhydrous ethanol were obtained from Chongqing Chuandong Chemical (Group) Co. (Chongqing, China). The calcium chloride anhydrous and 1,1,1,3,3,3-Hexafluoro-2-propanol (HFIP) (purity > 99.5%) were purchased from Aladdin Chemical Co., Ltd. (Shanghai, China). The Dulbecco’s modified Eagle medium (DMEM), fetal bovine serum (FBS), trypsin−EDTA, and penicillin-streptomycin were purchased from Gibco BRL, Rockville, MD, United States. The CCK-8 assay and Calcein/PI Cell Viability/Cytotoxicity Assay Kit were obtained from Beyotime Biotechnology companies (Shanghai, China).

### 4.2. Preparation of BP Nanosheets

#### 4.2.1. Preparation of SF Aqueous Solutions

To obtain SF, the Bombyx mori cocoons were first divided into 1 cm^2^ sheets and boiled twice for 30 min in 0.5% (w/w) Na_2_CO_3_ solution (liquid ratio 1:50) at 100 °C. The degummed cocoons were washed for three times and dried at 45 °C. Then, the dried SF was dissolved in a ternary solvent system of CaCl_2_/ C_2_H_5_OH/H_2_O (1:2:8 M ratio) at 75 °C (liquid ratio 1:10) for 2 h. After cooling down, the mixed solution of silk fibroin was centrifuged (5000 rpm, 5 mins), and then dialyzed using a semipermeable membrane (MWCO: 3.5–5 kDa) for 3 days. Finally, the regenerated SF sponge was prepared by freeze -drying [56,57]. Then, 10 mL of SF solution and 6.5 mL of 1 M chloroacetic acid were mixed on a magnetic stirrer (85-2A, Jintan Scientific Analytical Instruments Co., Ltd. (Jiangsu, China)) at a constant temperature of 25 °C for 1 h. The filamentous aggregates were removed from the SF aqueous solution through dialysis and centrifugation, and the clarified supernatant was then kept in reserve at 4 °C.

#### 4.2.2. Preparation of BP@SF

The BP@SF were prepared by ultrasound-assisted liquid exfoliation using bulk BP as the raw material and SF as the exfoliating agent. Firstly, 20 mg of BP bulk powder was dispersed in 20 mL of a 5 w/w % SF aqueous solution (SF:BP = 50:1). Nitrogen was added to the mixture, to separate the oxygen and prevent the oxidation of the BP, which was then sonicated for two hours in an ice bath, using an ultrasonic cell crusher (SCIENTZ-IID, Ningbo Xinzhi Biotechnology Co., Ltd. (Zhejiang, China)). To remove the unstripped BP, the solution was centrifuged for 30 min at 1500 rpm with a frozen high-speed centrifuge (TGL-20MS, Shanghai Xiang Yi Centrifuge Instruments Co., Ltd. (Shanghai, China)), then for another 30 min at 6000 rpm to obtain the BP@SF powder. Finally, an electric blast dryer (DHG-9245A, Shanghai Qiaoxin Scientific Instruments Co. (Shanghai, China)) was used to dry and store the prepared BP@SF.

### 4.3. Preparation of SF/PLGA/BP@SF Electrospun Composite Membranes

The 2–10% SF solution was prepared by referring to 4.2.1, and then reverse dialyzed with PEG (15–20%) for 1–2 days to remove water to increase the concentration and facilitate the freeze-drying to obtain the SF powder.

Four different concentrations of the BP@SF suspensions were obtained by ultrasonicating 0 mg, 1 mg, 2 mg, and 4 mg of BP@SF in 9 g of HFIP for 60 min, which were recorded as control, BP@SF-1, BP@SF-2, and BP@SF-4, respectively. The SF/PLGA spinning solution and the SF/PLGA/BP@SF spinning solutions of the different BP@SF concentrations were produced by magnetic stirring for 4 h at room temperature. The spinning was then carried out using an electrospinning machine (TL-Pro-BM, Shenzhen, China). A 21G metal needle was attached to a 10 mL syringe and each of the four groups of spinning solutions was injected into the syringe with the front end of the metal needle connected to the positive voltage and a grounded barrel with tin foil as a receiver device connected to the negative voltage. The spinning conditions were a receiving distance of 7.5 cm, a solution flow rate of 4 mL/h, a positive voltage of 20 kV, a negative voltage of −2 kV, a humidity of 60%, a temperature of 25–30 °C, and a drum speed of 600 rpm. The spinning time was approximately 2–3 h. The electrospun membranes were deposited on tin foil, which was then carefully peeled from the spinning drum and vacuum dried to remove the residual HFIP to obtain the electrospun membranes.

### 4.4. Characterizations of BP@SF

The morphologies of the BP@SF were observed by a Scanning Electron Microscope (SEM, JSM-6610, Eindhoven, The Netherlands) in a high vacuum with an acceleration voltage of 5–20 kV. The samples were previously coated with gold-palladium. In addition, the BP nanosheets’ morphology (high resolution) was photographed, using a transmission electron microscope (TEM) (JEOL JEM-F200, Japan), accelerating voltage 200 kV, and energy spectrum (JED-2300T) after drying. TEM procedure: A portion of the sample was dispersed in an aqueous solution and sonicated for two minutes. After that, a few drops of the dispersed liquid were added drop by drop to the copper mesh ltra-thin carbon Software programs ImageJ (1.51j8) and Digital Micrograph (3.7.4) were used to calculate the BP’s diameter and lattice spacing, respectively.

The particle size distribution of the BP@SF was tested with a nanoparticle size analyzer (Zetasizer NanoS90. Malvern City, UK). A total of 1 mg of BP@SF was dispersed uniformly in 1 mL of PBS. The solution was scanned 10 times.

FTIR spectroscopy (Bruker, Beijing, China) was performed to record the composition of the material and analyze the structure of the substance composition of the BP@SF. The scanning range was 4000–400 cm^−1^. A total of 1 mg of dried sample and 100 mg of KBr were ground well and pressed under a compressor to form transparent and homogeneous flakes, which were scanned by FTIR spectroscopy.

The X-ray diffraction spectra were recorded by the X-ray diffraction test system of Dandong Tongda Technology Co., Ltd. (Liaoning, China), using monochromatic Cu Kα1 ray radiation (λ = 1.54 nm). The angles ranged from 5° to 50° in steps of 0.02°.

Aqueous solutions of BP@SF at 0.1 mg/mL, 0.2 mg/mL and 0.4 mg/mL were prepared separately, with PBS as the control group. A total of 2 mL of the solutions were placed in a cuvette and the samples were detected by UV spectrophotometer (TU-1901. Beijing Pu-Analysis General Instrument Co. (Beijing, China)) in the 400–900 nm spectrum.

The aqueous BP@SF were irradiated with a near infrared laser (808 nm, 0.5 W/cm^2^) for 10 min in a 1 cm square cuvette. The temperature of the aqueous BP nanosheets solution was recorded by a digital thermometer (data represent means ± SD (*n* = 3)).

### 4.5. Characterizations of SF/PLGA/BP@SF Electrospun Composite Membranes

The morphologies of the BP@SF and electrospun membranes were observed by scanning electron microscopy (SEM, Phenom, JSM-6610, Eindhoven, Netherlands)) in high vacuum at accelerating voltages of 5–20 kV. The samples were previously coated with gold-palladium by magnetron sputtering. Nano Measure 1.2.5 software was used to determine the average diameter and diameter distribution of the fibers. The EDS elemental analysis test is based on SEM-scanned images for elemental content and distribution analysis.

The surface hydrophobicity of the electrospun membranes was tested repeatedly, with the SDC-300 optical contact angle tester. Briefly, the membranes were made 1 cm × 1 cm and glued to a glass sheet. Drops of water were placed on the membrane for 30 s before imaging at 25 °C (the test was repeated three times for each set of samples). The contact angle of the samples at 25 °C was assessed using a contact angle meter (OCA15EC, Datephysics, Fildestadt, Germany). Each sample was tested for tensile strength, using a universal material testing machine at room temperature with a humidity of 65%. The sample size was 30 mm × 10 mm and the thickness was approximately 0.006–0.15 mm. The effective tensile length of the electrospun membrane was 10 mm, using a 1 KN sensor and the tensile speed was 2 mm/min. The Young’s elasticity, fracture strength, and fracture elongation were averaged over five tests.

### 4.6. Photothermal Effect Measurement

To test its photothermal conversion performance, the samples were irradiated directly with 0, 0.5, 1, 1.5, 2 and 2.5 W. While the temperature rose to its maximum, photographs were taken and the elevated temperature was recorded. To test the stability of its photothermal conversion, a 50 mg sample of SF/PLGA/BP@SF-4 was placed in 50 μL of PBS and irradiated with 2.5 W of NIR for 10 min and then stopped. The temperature at which the solution rose and the time it took to cool naturally to room temperature was recorded. The above operation is repeated five times for cyclic testing.

### 4.7. Cell Culture

Mouse fibroblasts L929 and human hepatocellular carcinomas HeGp2 were acquired from the State Key Laboratory of Silkworm Genome, Southwest University. The cells were all cultured in DMEM high sugar medium, containing 10% fetal bovine serum, 100 μ/mL penicillin, and 100 μg/mL streptomycin, and placed in a 37 °C, 5% CO_2_ incubator.

### 4.8. Biocompatibility Test

The biocompatibility of the electrospun membranes on the L929 mouse fibroblasts was tested by a direct contact test. Staining was observed using a CCK-8 assay and LIVE/DEAD double fluorescent staining assay. The membranes of 6 mm diameter were previously irradiated in UV radiation for 1 h and subsequently placed in the bottom of each well of a 96-well plate pre-cultured with complete medium for 30 min. A total of 8 × 10^3^ cells/mL of the L929 cell suspension was subsequently added to each well and seeded at 37 °C and 5% CO_2_ for 3 days. The cells cultured in the wells without membranes were used as controls, and the SF/PLGA/BP@SF-1, SF/PLGA/BP@SF-2, and SF/PLGA/BP@SF-4 membranes were the experimental groups, with five replicate wells in each group. On days 1, 2, and 3 after the cell seeding, 250 μL DMEM medium and 25 μL CCK-8 assay were added to each well and incubated at 37 °C with 5% CO_2_ for 1 h. The absorbance at 450 nm was then measured with a microplate reader (Synergy H1, BioTek, Beijing Representative Office; USA). Three days later, 0.1 μL Calcein AM, 0.1 μL Propidium Iodide (PI), and 0.1 mL Test buffers were added to the plates, incubated for 30 min and the cells were observed by fluorescence microscopy (BX53, Olympus, Shanghai, China).

### 4.9. In Vitro Anti-Tumor Performance Testing

In this experiment, the inhibitory properties of the cancer cells were evaluated by co-culture of HeGp2 cells on nanofiber membranes, stained with CCK-8 assay, and the LIVE/DEAD double fluorescent staining assay. After two passages, 96-well plates with 1 × 10^4^ cells per well were plated and incubated in a cell culture incubator (37 °C, 5% CO_2_) for 24 h. The nanofiber membranes were sterilized by light treatment for 30 min and then added to the 96-well plates for 4 h, followed by 808 nm NIR light irradiation at 2.5 W/cm^2^ for 10 min. The OD values were measured by CCK-8 staining at 24 h and 48 h after the light treatment. The LIVE/DEAD double fluorescent staining assay was stained at the 48th hour and the staining status was observed by fluorescence microscope. Statistical plotting was completed using OriginPro software.

### 4.10. Statistical Analysis

All of the quantitative data were expressed as Mean ± Standard Deviation. All of the analyses of variance (ANOVA one-way, OriginPro 2021b (64-bit, 9.8.5.201, OriginLab Corporation) were performed through post-hoc Tukey tests, to test the significance of the results. A *p*-value < 0.05 indicated statistical significance: * is *p* < 0.05, ** is *p* < 0.01, and *** is *p* < 0.001.

## 5. Conclusions

In conclusion, the BP@SF with good water dispersion stability was prepared by ultrasound-assisted liquid exfoliation, using SF as an effective exfoliator. Based on the excellent solution processability of the BP@SF, SF/PLGA/BP@SF electrospun membranes were prepared, together with SF and PLGA, by electrospinning. The fabricated SF/PLGA/BP@SF electrospun membranes exhibited excellent mechanical properties, good biocompatibility, and controllable photothermal conversion properties under NIR for effectively killing cancer cells, which gives them great potential in the treatment of cancer. However, there were also some challenges in this study. For example, only in vitro cellular experiments and cancer cell ablation experiments were performed. In vivo animal studies are required for further evaluation.

## Figures and Tables

**Figure 1 molecules-27-04563-f001:**
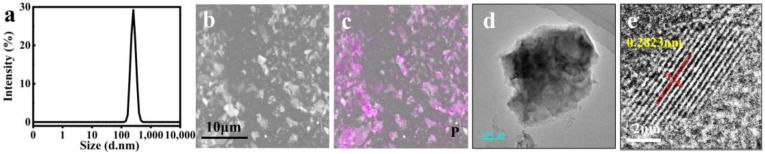
Morphologies of BP@SF nanosheets. (**a**) Particle size distribution (PDI = 0.688); (**b**) SEM. Scale bar of 10 nm; (**c**) EDS. Phosphorus distribution, scale bar of 10 nm; (**d**) TEM. Scale bar of 200 nm; (**e**) selected area electron diffraction. Scale bar of 2 nm.

**Figure 2 molecules-27-04563-f002:**
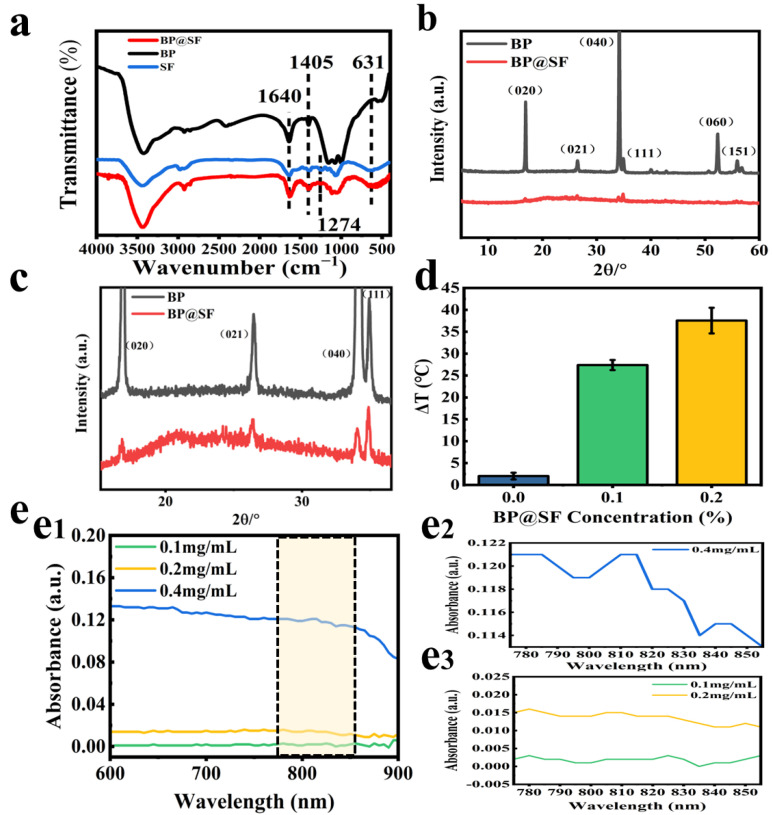
Characterization of BP@SF. (**a**) XRD from 5 to 60°; (**b**) XRD from 15 to 36°; (**c**) FTIR; (**d**) Heating effect of BP@SF aqueous solutions; (**e**) UV-Vis-NIR absorption spectra of thin-layer BP@SF nanosheets; (**e_1_**) wavelength from 600 to 900 nm; (**e_2_**) The absorbance of 0.4 mg/mL BP@SF from 775 to 855 nm; (**e_3_**) The absorbance of 0.1 mg/mL BP@SF and 0.2 mg/mL BP@SF from 775 to 855 nm.

**Figure 3 molecules-27-04563-f003:**
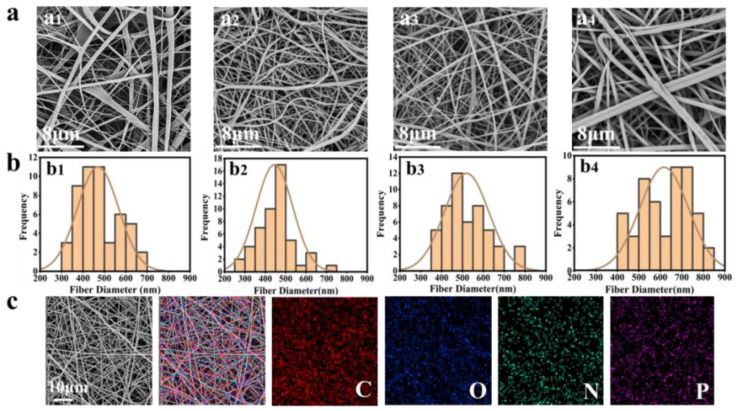
Morphology and characterization of the electrospun membranes. (**a**,**b**) SEM images (10,000×) and diameter distribution of 50 fibers in electrospun membranes of different components. (**a_1_**,**b_1_**) SF/PLGA; (**a_2_**,**b_2_**) SF/PLGA/BP@SF-1; (**a_3_**,**b_3_**) SF/PLGA/BP@SF-2; (**a_4_**,**b_4_**) SF/PLGA/BP@SF-4; (**c**) Elemental scanning of carbon, nitrogen, oxygen and phosphorus in SF/PLGA/BP@SF-4 electrospun membrane. Separate detection of elemental distribution of carbon, oxygen, nitrogen and phosphorus.

**Figure 4 molecules-27-04563-f004:**
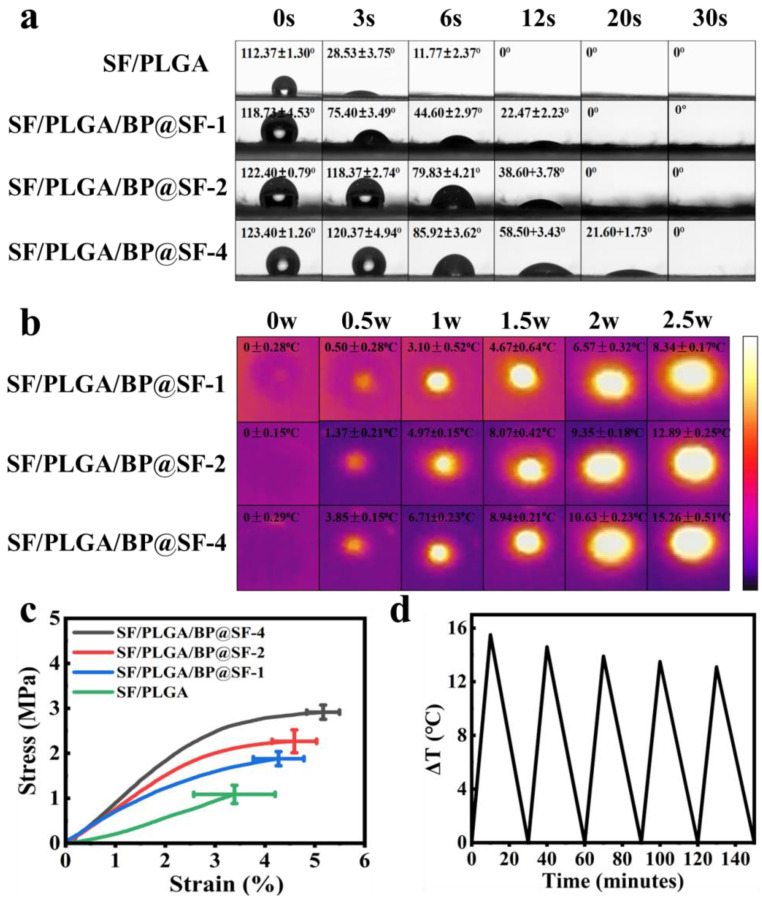
(**a**) The contact angles of the electrospun membranes; (**b**) Photothermal effect; (**c**) Tensile properties; (**d**) the SF/PLGA/BP@SF-4 cycle test of heating under near-infrared irradiation with 2.5 W power.

**Figure 5 molecules-27-04563-f005:**
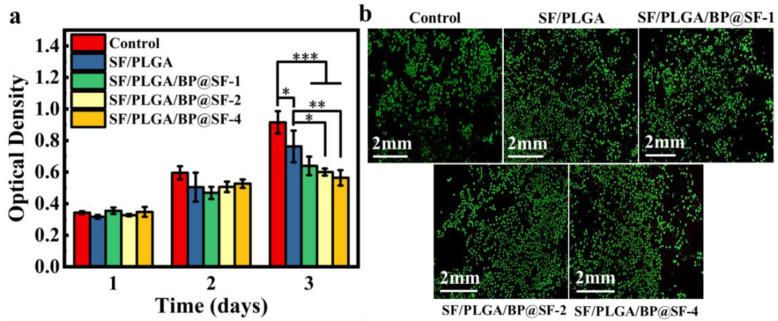
Biocompatibility of electrospinning membrane. (**a**) Evaluation of cell viability by CCK-8 assay; (**b**) LIVE/DEAD cell staining of L929 cells on the third day of culture. * *P* < 0.05, ** *P* < 0.01, *** *P* < 0.001, data represent means ± SD.

**Figure 6 molecules-27-04563-f006:**
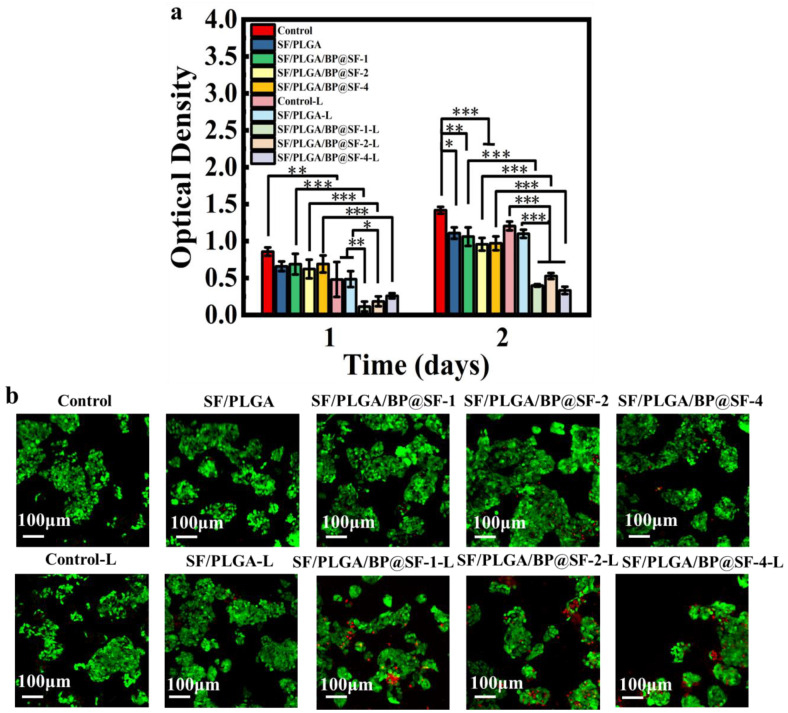
In vitro anti-tumor properties of the electrospun membranes. (**a**) Evaluation of cell viability by CCK-8 assay; (**b**) LIVE/DEAD cell staining of HepG2 cells on the second day of culturing. * *P* < 0.05, ** *P* < 0.01, *** *P* < 0.001, data represent means ± SD.

**Figure 7 molecules-27-04563-f007:**
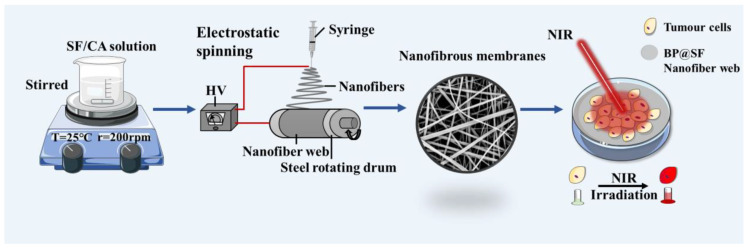
Schematic diagram of sample preparation.

**Table 1 molecules-27-04563-t001:** P elemental analysis of electrospun membranes.

Sample	Atomic Conc.	Weight Conc.
SF/PLGA	0.06%	0.13%
SF/PLGA/BP@SF-1	0.06%	0.14%
SF/PLGA/BP@SF-2	0.08%	0.18%
SF/PLGA/BP@SF-4	0.11%	0.25%

**Table 2 molecules-27-04563-t002:** EDS elemental analysis of SF/PLGA/BP@SF-4 membrane.

Element Number	Element Symbol	Element Name	Atomic Conc.	Weight Conc.
6	C	Carbon	43.45%	37.82%
8	O	Oxygen	32.02%	37.13%
7	N	Nitrogen	24.42%	24.79%
15	P	Phosphorus	0.11%	0.25%

## Data Availability

Not applicable.

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
