# Peer review of "Electrospun Silk Fibroin/Polylactic-co-glycolic Acid/Black Phosphorus Nanosheets Nanofibrous Membrane with Photothermal Therapy Potential for Cancer"

_molecules, 2022, doi:10.3390/molecules27144563_

Round 1
Reviewer 1 Report
The manuscript “Electrospun Silk Fibroin/Poly(lactic-co-glycolic acid)/Black Phosphorus Nanosheets Nanofibrous Membrane with Photothermal Therapy Potential for Cancer” reports the exfoliation of black phosphorus employing silk fibroin with the subsequent generation of fibrous membranes combining with a poly(lactic-co-glycolic acid) copolymer by electrohydrodynamic procedures. The generated mats were characterized by employing some typical techniques. Some materials were tried from the point of view of the photothermal activity carrying out also in vitro assays in presence of HepG2 cancerous cells. Despite the research looking very interesting, I am attaching some comments and suggestions which I would like to be considered by the authors.
Comments.
1- Introduction. Please, improve the description of the electrospinning technique for the obtention of mats focussing on biological applications.
2- Figure 1 is not adequate for this part of the manuscript.
3- About the diameter of BP@SF. If are not spherical or similar in shape, the DLS technique is not useful. Please, incorporate some electronic microscopy (SEM/TEM) or AFM images.
4- Fig. 2D. Temperature is normalized? I recommend expressing it as ΔT (Final temperature – Initial temperature).
5- Section 2.2. Preparation and characterization of BP@SF electrospun membranes. Why the authors did not express the amount of incorporated BP as a percentage of the SF concentration?
6- I am curious about why did the authors choose high humidity percentage. The flow rates during the electrohydrodynamic process seems also high. Is it due to any particular reason?
7- SF/PLGA/BP@SF diameters: This is not clear in the analysis. Please, improve this part by considering also the results of Fig. 3b.
8- Why EDS analysis is only carried out for one of the samples?
9- "Hydrophobicity is one of the important properties of the electrospun fibrous membrane". Reference is missing.
10- Section 2.4. Need to be explained in a deep form.
11- "Photothermal properties are crucial for PTT". References are missing?
12- Fig 4d. Express the temperature as changes. Similar comment to point #4.
13- “In vitro” and “in vivo” should be in italic.
14- In vitro anti-tumor properties section and Fig 6. The study is not clear from the point of view of time. Besides, why only one of the systems is assayed?
15- Why did the authors use a rotating drum and selected 600 rpm as the speed?
16- How the mat thicknesses were measured? Explain and report an average indicating the n.
17- Did the authors evaluate the degradation of the obtained materials? It could be an interesting point to explore.
18- Conclusions should be improved.
Author Response
Point 1: Introduction. Please, improve the description of the electrospinning technique for the obtention of mats focussing on biological applications.
A: Thanks for your suggestion on electrospinning.
Here are the details of revision.
One of the largest fields of application of electrospun materials is the owing to the large surface area, controllable surface configuration, well-modified surface, complex pore structure and good biocompatibility [25-31], electrospun membranes are wildly used in biomedical field, including tissue engineering, drug delivery and wound healing. (Line 94 to 97 in revised manuscript).
Reference:
[25] Zhang, Z., et al., Sandwich-Like Fibers/Sponge Composite Combining Chemotherapy and Hemostasis for Efficient Postoperative Prevention of Tumor Recurrence and Metastasis. Advanced Materials, 2018. 30(49): p. 1803217.
[26] Sun, J., et al., Synergistic Photodynamic and Photothermal Antibacterial Nanocomposite Membrane Triggered by Single NIR Light Source. ACS Applied Materials & Interfaces, 2019. 11(30): p. 26581-26589.
[27] Mauro, N., et al., A self-sterilizing fluorescent nanocomposite as versatile material with broad-spectrum antibiofilm features. Materials Science and Engineering: C, 2020. 117: p. 111308.
[28] Wang, X., et al., Electrospun Micropatterned Nanocomposites Incorporated with Cu2S Nanoflowers for Skin Tumor Therapy and Wound Healing. ACS Nano, 2017. 11(11): p. 11337-11349.
[29] Lee, Y.B., et al., Ternary nanofiber matrices composed of PCL/black phosphorus/collagen to enhance osteodifferentiation. Journal of Industrial and Engineering Chemistry, 2019. 80: p. 802-810.
[30] Tolba, M.F., et al., Editorial: Immunotherapy as an Evolving Approach for the Treatment of Breast Cancer. Frontiers in Oncology, 2021. 11.
[31] Lucie Depeigne et al. Electrospun Biomaterials’ Applications and Processing. Journal of Biomimetics, Biomaterials and Biomedical Engineering. 2021.49(91-100), 2296-9845.
Point 2: Figure 1 is not adequate for this part of the manuscript.
A: The order has been changed, and it is indicated in the manuscript (Fig. 7).
Point 3: About the diameter of BP@SF. If are not spherical or similar in shape, the DLS technique is not useful. Please, incorporate some electronic microscopy (SEM/TEM) or AFM images.
A: The description has been revised, and SEM and TEM image have been introduced to characterize the exfoliated BP@SF, which are marked in the manuscript.
Here are the details.
BP@SF sheets were prepared by ultrasound-assisted liquid exfoliation using SF as an effective exfoliator. With a unique hydrophilic-hydrophobic structure and abundant carboxyl groups, SF can facilitate the interfacial bonding with 2D nanomaterials [35, 36]. On this basis, during the exfoliation process, SF molecules stably bound to the BP crystal surface through strong hydrophobic interactions. Meanwhile, their hydrophilic regions are exposed to water to stabilize the exfoliated BP nanosheets and prevent re-agglomeration. Ultrathin BP@SF sheets were obtained after sonication of BP powders in aqueous SF solution for 2 h. The size of BP@SF was from 200 to 400 nm (PDI=0.688) (Fig. 1a). SEM observation (Fig. 1b) and EDS analysis (Fig. 1c) showed that BP@SF were well distributed. The exfoliated BP@SF exhibited lamellar appearance and the lattice spacing of the was 0.2823 nm (Fig. 1d and Fig. 1e), which was consistent with the study reported [20]. (Line 116 to127 in revised manuscript)
Figure 1. Morphologies of BP@SF nanosheets. (a) Particle size distribution (PDI=0.688). (b) SEM. Scale bar of 10 nm. (c) EDS. Scale bar of 10 nm. (d) TEM. Scale bar of 200 nm. (e) Selected area electron diffraction. Scale bar of 2 nm.
Reference:
[20] Xiao-Wei Huang et al. Water-Based Black Phosphorus Hybrid Nanosheets as a Moldable Platform for Wound Healing Applications. ACS APPLIED MATERIALS&INTERFACES[J] 2018, 10, 35495−35502.
[35] Huang, X., et al., Silk fibroin-assisted exfoliation and functionalization of transition metal dichalcogenide nanosheets for antibacterial wound dressings. Nanoscale, 2017. 9(44): p. 17193-17198.
[36] Grant, A.M., et al., Silk Fibroin–Substrate Interactions at Heterogeneous Nanocomposite Interfaces. Advanced Functional Materials, 2016. 26(35): p. 6380-6392.
Point 4: Fig. 2D. Temperature is normalized? I recommend expressing it as ΔT (Final temperature – Initial temperature).
A: The image has been revised according to this suggestion and the revised figure (Fig. 2d in manuscript) is shown as follows.
Figure 2. (d) Heating effect of BP@SF aqueous solutions.
Point 5: Section 2.2. Preparation and characterization of BP@SF electrospun membranes. Why the authors did not express the amount of incorporated BP as a percentage of the SF concentration?
A: The exfoliated BP@SF is a composite that the amount of incorporated BP cannot be calculated according to our knowledge. We can just measure the weight of BP@SF and use the BP@SF weight for study.
Point 6: I am curious about why did the authors choose high humidity percentage. The flow rates during the electrohydrodynamic process seems also high. Is it due to any particular reason?
A: This condition for electrospinning of PLGA/SF was verified in our previous study [1], since the spun fibers would have some beads or have the uneven appearance if the condition was varied.
Reference:
[1] Peng Y, Ma Y, et al. Electrospun PLGA/SF/artemisinin composite nanofibrous membranes for wound dressing[J]. International Journal of Biological Macromolecules. 183 (2021) 68–78.
Point 7: SF/PLGA/BP@SF diameters: This is not clear in the analysis. Please, improve this part by considering also the results of Fig. 3b.
A: This part for fiber diameter has been revised in manuscript.
Here are the details.
SEM observation (Fig. 3a) showed that the electrospun SF/PLGA fibers have the smooth appearance and the average fiber diameter was approximately 470.40 ± 96.17 nm, while the electrospun fibers with loaded BP@SF had the relatively rough surface and their average fiber diameter varies (Fig. 3b). The fiber diameter of SF/PLGA/BP@SF-1 ranged from 200 nm to 800 nm. With further increasing of loaded BP@SF content, the fibers ranged from 300 nm to 900 nm and the average diameter of the fibers significantly increased from 444.60 ± 90.20 nm (SF/PLGA/BP@SF-1) to 607.60 ± 116.93 nm (SF/PLGA/BP@SF-4) (Fig. 3b). (Line 171 to 178 in revised manuscript)
Point 8: Why EDS analysis is only carried out for one of the samples?
A: EDS tests were carried out to identify if BP@SF was evenly loaded on fibrous membrane through electrospinning. According to this suggestion, the P content of four electrospun membranes have been identified and the results are listed in Table 1 in revised manuscript. And the description is also added.
Here are the details.
EDS analysis indicated the content of P element was 0.13%(Table. 1), which was from SF protein. With loaded BP@SF, P element content increased and reached up to 0.25% for SF/PLGA/BP@SF-4 (Fig. 3c and Table. 2). which identified the BP@SF was evenly loaded on fibrous membrane through electrospinning. (Line 187 to 191 in revised manuscript)
Table 1. The content of P element on electrospun membranes.
Sample
|
Atomic Conc.
|
Weight Conc.
|
SF/PLGA |
0.06% |
0.13% |
SF/PLGA/BP@SF-1 |
0.06% |
0.14% |
SF/PLGA/BP@SF-2 |
0.08% |
0.18% |
SF/PLGA/BP@SF-4 |
0.11% |
0.25% |
Point 9: "Hydrophobicity is one of the important properties of the electrospun fibrous membrane". Reference is missing.
A: The references have been added to the revised version after careful review of the literature.
Hydrophobicity is one of the important properties of electrospun fibrous membrane [45]. (Line 207 to 208 in revised manuscript)
Reference:
[45] Rafael S. Kurusu. Surface modification to control the water wettability of electrospun mats. International Materials Reviews. 64(2019)249-287.
Point 10: Section 2.4. Need to be explained in a deep form.
A: I appreciate your thoughtful advice. Here are my revisions.
Appropriate mechanical properties of electrospun membrane are necessary in the application [47,48]. Therefore, the effect of BP@SF on the mechanical properties of electrospun membrane were investigated using a tensile testing machine (Fig. 4c). Among these electrospun membranes, SF/PLGA membrane had the lowest stress, strain and Young's modulus, which were 1.51 ± 0.89 MPa, 229 ± 3.3% MPa and 254.83 ± 3.37 MPa, respectively. Compared to the SF/PLGA electrospun membrane, the breaking strength and elongation at break of the electrospun membrane increased correspondingly with the increasing of BP@SF content in the electrospun membrane. In particular, the stress, strain and Young's modulus of SF/PLGA/BP@SF-4 membrane enhanced to 2.92 ± 0.32 MPa, 51 ± 2.3% and 104.94 ± 2.15 MPa, respectively. This may attribute to the good mechanical properties of the BP nanosheets.(Line 225 to 236 in revised manuscript)
Reference:
[47] Dan Tian et al. Control of Macromolecule Chains Structure in a Nanofiber. Polymers. 2020, 12(10), 2305.
[48] Saad Nauman et al. Post Processing Strategies for the Enhancement of Mechanical Properties of ENMs (Electrospun Nanofibrous Membranes): A Review. Membranes. 2021, 11(1), 39.
[49] Mu, X., J. Wang and M. Sun, Two-dimensional black phosphorus: physical properties and applications. Materials Today Physics, 2019. 8: p. 92-111.
Point 11: "Photothermal properties are crucial for PTT". References are missing?
A: The references have been added to the revised version after careful review of the literature. Here are my revisions: For PTT, photothermal characteristics are essential [50].(Line 238 in the manuscript)
Reference:
[50] Jing Sun et al. Synergistic Photodynamic and Photothermal Antibacterial Nanocomposite Membrane Triggered by Single NIR Light Source. ACS Appl. Mater. Interfaces 2019, 11, 30, 26581–26589.
Point 12: Fig 4d. Express the temperature as changes. Similar comment to point #4.
A: The image has been revised according to this suggestion (Fig. 4d in manuscript).
Point 13: “In vitro” and “in vivo” should be in italic.
A: We sincerely apologized for our careless errors. I appreciate you reminding me. We carefully reviewed the manuscript and made the necessary corrections.
Point 14: In vitro anti-tumor properties section and Fig 6. The study is not clear from the point of view of time. Besides, why only one of the systems is assayed?
A: We revise the manuscript according to your suggestions. All samples were included in anti-cancer cell experiments. Here are the revisions.
Human hepatocarcinoma cell line (HepG2) was inoculated onto the membranes and the anti-tumour effect of SF/PLGA/BP@SF membranes under the photothermal effect was investigated in vitro (Fig. 6a). On the first day, the cell viability shows no significant difference among the five groups without NIR treatment. After treated by NIR (808 nm, 2.5 W/cm2 for 10 mins), the cell viability for SF/PLGA/BP@SF-1-L, SF/PLGA/BP@SF-2-L and SF/PLGA/BP@SF-4-L dramatically decrease, respectively, while the cell viability of SF/PLGA-L group has no significant reduction. It was evident that the BP@SF on fibrous membrane convert the NIR to heat, which inhibit the cancer cell growth and proliferation, leading to the lower cell viability. There has a similar findings for cell viability on the second day. Further, the LIVE/DEAD cell staining was conducted to observe the cell stadium after treating by NIR (Fig. 6b). The cells of control and SF/PLGA group show green fluorescence and almost all cells are alive, while a great numbers of dead cells for SF/PLGA/BP@SF-1-L group, SF/PLGA/BP@SF-2-L group and SF/PLGA/BP@SF-4-L group are found (shown in red), indicating that the photothermal effect of these groups can significantly kill the HeGp2 cancer cells. These results demonstrate that the SF/PLGA/BP@SF membranes has excellent and controllable photothermal property, which has a good potential for application in cancer therapy. (Line 280 to 296 in the manuscript)
Point 15: Why did the authors use a rotating drum and selected 600 rpm as the speed?
A: The condition was based on our previous studies [1].
Reference:
[1] Peng Y, Ma Y, et al. Electrospun PLGA/SF/artemisinin composite nanofibrous membranes for wound dressing[J]. International Journal of Biological Macromolecules. 183 (2021) 68–78.
Point 16: How the mat thicknesses were measured? Explain and report an average indicating the n.
A: Thank you for the insightful thoughts. Vernier calipers were used to measure the electrospun membranes' thickness according to the reported studies [1], and five different points of each membrane was measured, and the data were averaged.
Reference:
[1] Fouad Junior Maksoud. Et al. Electrospun waterproof breathable membrane with a high level of aerosol filtration. Applied polymer. 2018, 135(2), 45660
Point 17: Did the authors evaluate the degradation of the obtained materials? It could be an interesting point to explore.
A: We are very appreciative of the reviewers' ideas. Since electrospun membranes are applied in vitro and have the advantage of being readily replaceable, degradability is not the focus in this study
Point 18: Conclusions should be improved.
A: The conclusion part has been revised. Here are my revisions.
In conclusion, BP@SF with good water dispersion stability was prepared by ultrasound-assisted liquid exfoliation using SF as an effective exfoliator. Based on the excellent solution processability of BP@SF, SF/PLGA/BP@SF electrospun membranes were prepared together with SF and PLGA by electrospinning. The fabricated SF/PLGA/BP@SF electrospun membranes exhibit the excellent mechanical properties, good biocompatibility and controllable photothermal conversion properties under NIR for effectively killing cancer cells, which make them a great potential in the treatment of cancer. However, there are some also challenges in this study. For example, only in vitro cellular experiments and cancer cell ablation experiments have been performed. In vivo animal study is required for further evaluation. (Line 518 to 527 in the manuscript)
Reviewer 2 Report
This paper is about the preparation, characterization, and in-vitro evaluation against HepG2 cancer cells of the electrospun nanofiber conjugates of silk fibroin with black phosphorus and PLGA for anticancer photothermal therapy applications. The research has demonstrated the successful achievement of the material using FTIR, SEM, and XRD. It also showed significant photothermal conversion by NIR irradiation and photothermal hyperthermia reduction of HepG2 cancer cells proliferation, using the LIVE/DEAD double fluorescent staining assay. Biocompatibility on L929 mouse fibroblasts was observed using the CCK-8 assay. The hydrophilicity and tensile properties including the contact angles, stress, strain, and Young's modulus, respectively, needed for the contemplated applications were measured. The diameters of the silk fibers were measured.
The approach used by the researchers to prepare the material is sound. The characterization is adequate. Measurement of the hydrophilicity and tensile properties and diameters of the silk fibers are very sound. The scanning electron micrographs support the structural determinations. The photothermal conversions presented in bar graphs could have been more convincingly shown in photothermal curves showing the rise in temperature against time upon NIR irradiation. The Cycle test of the photothermal effect for SF/PLGA/BP@SF-4 under NIR with 2.5 W power is either not adequately described in the paper or is lost under the poor language usage. Hence my main issue with the paper is English language usage. I started off listing the language corrections and then gave up as there are too many of them. I cannot recommend publishing this paper with the language as it is.
In this paper, it is not really going to assist to do point-by-point corrections. An English language usage specialist can help improve the manuscript. If necessary, one can read the paper again, thereafter and change the recommendation. Therefore, I recommend it for publication only subject to English language editing.
Author Response
Point 1: In this paper, it is not really going to assist to do point-by-point corrections. An English language usage specialist can help improve the manuscript. If necessary, one can read the paper again, thereafter and change the recommendation. Therefore, I recommend it for publication only subject to English language editing.
A: We sincerely apologise for any difficulties that these errors in this manuscript may have caused you when reading it. The manuscript has undergone extensive grammar and spelling checks and these errors have been corrected. We therefore hope that it will meet the requirements of the journal.
Reviewer 3 Report
In this work, the authors report a new exfoliation approach for the preparation of silk fibroin-coated black phosphorous quantum dots, its formulation in electrospun nanofibers and their preliminary applications towards the photothermal treatment of cancer in vitro. Even though promising results are presented, some important issues need to be addressed before considering the manuscript for publications in the Molecules journal.
Materials and Methods:
· The overall work would be more understandable if the Materials and methods section is moved right after the Introduction section, and before the Results and Discussion section.
· The synthetic procedure for the preparation of the stable aqueous solution of silk fibroin in water needs to be clearer. In particular, in paragraph 4.2.1. “Preparation of SF aqueous solutions”:
- It should be specified which purification approach was employed to remove Na2CO3 used for degumming SF and how it was desiccated, since the following sentence starts with “27 g of desiccated cocoons”.
- The reaction conditions of the treatment with lithium bromide are not clearly specified. Was the reaction performed at room temperature? The same comment applies to the carboxymethylation of SF with chloroacetic acid, for which the temperature was not specified.
· In paragraph 4.2.2. “Preparation of BP@SF”:
- The concentration of SF employed for the preparation of BP@SF should be specified.
- The sentence “The mixture was filled with nitrogen” is not clear and needs to be clarified. Does that mean that the oxygen is removed from solution via nitrogen bubbling?
Results and Discussion:
· My main concern relates to the missing discussion and characterization of carboxymethylated silk fibroin. The reaction of SF with chloroacetic acid, reported in the Materials and Methods section only, represent an extensive chemical modification of the silk fibroin that changes drastically its surface properties. The authors state that “with a unique hydrophilic-hydrophobic structure and abundant carboxyl groups, SF can facilitate the interfacial bonding with 2D nanomaterials” (lines 113-114) but this is true only for carboxymethylated SF, since unmodified SF is characterized by a very low carboxyl group content. Literature data (DOI:10.1021/ja01619a066) reveal that the most abundant aminoacids in the SF protein sequence are glycine (43%), alanine (30%) and serine (12%) while the COOH-containing aminoacids are present in very low percentages. Therefore, the authors should demonstrate that the chemical functionalization with chloroacetic acid is required for exfoliating BP into nanosheets by testing non-functionalized SF. In addition, the authors contradict themselves when they state that “during the exfoliation process, SF molecules stably bound to the BP crystal surface through strong hydrophobic interactions” (lines 115-116). The authors should clarify this point: is the chemical modification with chloroacetic acid required to bind BP due to the hydrophilic interaction between COOH groups and the BP surface or such modification is performed for improving water dispersibility while the hydrophobic portions of SF are responsible for the interaction with BP surface that led to its exfoliation? In any case, the carboxymethyl-modified SF should be characterized and discussed as a synthetic material in terms of the degree of functionalization and surface zeta potential. All these points were already discussed in previously published papers by different authors such as in DOI: 10.1039/c7nr06807g.
· In line 119, the PDI of the particle size analysis should be reported in order to allow for the evaluation of the broadness of the size distribution.
· In Figure 2c, the XRD spectrum of BP@SF in not clearly visible. The spectrum should be scaled up in order to allow for easier identification of the characteristic reflection peaks for BP since this data are not representative of the outcome of the synthesis if presented in the current form.
· The optical properties of the prepared BP@SF should be characterized by UV-VIS spectroscopy, in order to justify the irradiation at 808 nm for photothermal experiment. Are the nanosheets absorbing light throughout all the visible spectrum? Do they have significant absorption at 808 nm? The same can be said for the electrospun fibers, for which the optical transparency throughout the visible and NIR region of the electromagnetic spectrum should be analyzed.
· In Figure 2d, the label of the y-axis should be “Temperature Increase” rather than just “temperature”
· The chemical composition of BP@SF and of the electrospun fibers should be investigated with more precise tools than only EDS performed on SEM, which give semi-quantitative results. Other approaches for the quantification of the total phosphorous content of the preparations, such as ICP-AES or ICP-MS, should be employed to confirm the EDS results.
· The BP nanosheets should be characterized by TEM in order to assess crystallinity and thickness of the nanosheets.
· In Figure 3b, how many fibers were measured for making the size distribution histograms?
Author Response
Materials and Methods:
Point 1: The overall work would be more understandable if the Materials and methods section is moved right after the Introduction section, and before the Results and Discussion section.
A: I appreciate the reviewer's comments, but this writing order seems to be a format requirement for molecules journal.
Point 2: The synthetic procedure for the preparation of the stable aqueous solution of silk fibroin in water needs to be clearer. In particular, in paragraph 4.2.1. “Preparation of SF aqueous solutions” and It should be specified which purification approach was employed to remove Na2CO3 used for degumming SF and how it was desiccated, since the following sentence starts with “27 g of desiccated cocoons”. In addition,The reaction conditions of the treatment with lithium bromide are not clearly specified. Was the reaction performed at room temperature? The same comment applies to the carboxymethylation of SF with chloroacetic acid, for which the temperature was not specified.
A: The modification has been performed. The detailed revision is as follows.
To obtain SF, Bombyx mori cocoons were first divided into 1 cm2 sheets and boiled twice for 30 minutes in 0.5% (w/w) Na2CO3 solution (liquid ratio 1:50) at 100 °C. The degummed cocoons were washed for three times and dried at 45 °C. Then, the dried SF was dissolved in a ternary solvent system of CaCl2/ C2H5OH/H2O (1:2:8 M ratio) at 75 °C (liquid ratio 1:10) for 2 h. After cooling down, the mixed solution of silk fibroin was centrifuged (5000 rpm, 5mins) and then dialyzed using a semipermeable membrane (MWCO: 3.5–5 kDa) for 3 days. Finally, the regenerated SF sponge was prepared by freeze-drying [50, 51]. Then, 10 mL of SF solution and 6.5 mL of 1 M chloroacetic acid were mixed on a magnetic stirrer (85-2A, Jintan Scientific Analytical Instruments Co., Ltd.) at a constant temperature of 25 °C for 1 hour. The filamentous aggregates were removed from the SF aqueous solution through dialysis and centrifugation, and the clarified supernatant was then kept in reserve at 4 °C. (Line 384 to 395 in the manuscript)
Point 3: The concentration of SF employed for the preparation of BP@SF should be specified. In addition, the sentence “The mixture was filled with nitrogen” is not clear and needs to be clarified. Does that mean that the oxygen is removed from solution via nitrogen bubbling?
A: Thank you for the insightful thoughts. In the updated document, we have added the SF concentration. The reviewer's speculation is also accurate. To eliminate oxygen interference, the mixture is filled with nitrogen. I therefore changed the manuscript as the follows.
BP@SF were prepared by ultrasound-assisted liquid exfoliation using bulk BP as the raw material and SF as the exfoliating agent. Firstly, 20 mg of BP bulk powder was dispersed in 20 mL of a 5 w/w% SF aqueous solution (SF:BP=50:1). Nitrogen was added to the mixture to separate the oxygen and prevent the oxidation of the BP, which was then sonicated for two hours in an ice bath using an ultrasonic cell crusher (SCIENTZ-IID, Ningbo Xinzhi Biotechnology Co., Ltd.). To remove the unstripped BP, the solution was centrifuged for 30 minutes at 1500 rpm with a frozen high-speed centrifuge (TGL-20MS, Shanghai Xiang Yi Centrifuge Instruments Co., Ltd.), then for another 30 minutes at 6000 rpm to obtain the BP@SF powder. Finally, an electric blast dryer (DHG-9245A, Shanghai Qiaoxin Scientific Instruments Co.) was used to dry and store the prepared BP@SF. (Line 397 to 406 in the manuscript)
Point 4: My main concern relates to the missing discussion and characterization of carboxymethylated silk fibroin. The reaction of SF with chloroacetic acid, reported in the Materials and Methods section only, represent an extensive chemical modification of the silk fibroin that changes drastically its surface properties. The authors state that “with a unique hydrophilic-hydrophobic structure and abundant carboxyl groups, SF can facilitate the interfacial bonding with 2D nanomaterials” (lines 113-114) but this is true only for carboxymethylated SF, since unmodified SF is characterized by a very low carboxyl group content. Literature data (DOI:10.1021/ja01619a066) reveal that the most abundant aminoacids in the SF protein sequence are glycine (43%), alanine (30%) and serine (12%) while the COOH-containing aminoacids are present in very low percentages. Therefore, the authors should demonstrate that the chemical functionalization with chloroacetic acid is required for exfoliating BP into nanosheets by testing non-functionalized SF. In addition, the authors contradict themselves when they state that “during the exfoliation process, SF molecules stably bound to the BP crystal surface through strong hydrophobic interactions” (lines 115-116). The authors should clarify this point: is the chemical modification with chloroacetic acid required to bind BP due to the hydrophilic interaction between COOH groups and the BP surface or such modification is performed for improving water dispersibility while the hydrophobic portions of SF are responsible for the interaction with BP surface that led to its exfoliation? In any case, the carboxymethyl-modified SF should be characterized and discussed as a synthetic material in terms of the degree of functionalization and surface zeta potential. All these points were already discussed in previously published papers by different authors such as in DOI: 10.1039/c7nr06807g.
A: Thank you very much for the detailed explanation. The process of BP stripping in this paper was based on the study reported by Huang et al [20]. The chloroacetic acid carboxyl functionalisation was used to modify the SF, and then mixing solution was used to exfoliate the bulk BP. The characterisation of the chloroacetic acid-modified filamentous protein was not reported in the study. The reported mechanism of exfoliation is that the hydrophobic portion of SF and the enhanced carboxyl group can bind to the surface of the BP nanosheets. In addition, the hydrophilic portion of the SF was exposed to water, which might stabilize the exfoliated BP nanosheets and prevent re-agglomeration. Here, we focus on the inhibitory effect of electrospun membranes containing BP@SF on cancer cells under NIR irradiation, thus, the detailed mechanism for exfoliation is not included in this study.
Reference:
[20] Huang, X., et al., Water-Based Black Phosphorus Hybrid Nanosheets as a Moldable Platform for Wound Healing Applications. ACS Applied Materials & Interfaces, 2018. 10(41): p. 35495-35502.
Point 5: In line 119, the PDI of the particle size analysis should be reported in order to allow for the evaluation of the broadness of the size distribution.
A: The mistakes have been revised, which are marked in the manuscript.The detailed revision is as follows.
The size of BP@SF was from 200 to 400 nm (PDI=0.688) (Fig. 1a). (Line 123 in the manuscript)
Point 6: In Figure 2c, the XRD spectrum of BP@SF in not clearly visible. The spectrum should be scaled up in order to allow for easier identification of the characteristic reflection peaks for BP since this data are not representative of the outcome of the synthesis if presented in the current form.
A: The image has been revised according to this suggestion (Fig. 2 in manuscript).
Point 7: The optical properties of the prepared BP@SF should be characterized by UV-VIS spectroscopy, in order to justify the irradiation at 808 nm for photothermal experiment. Are the nanosheets absorbing light throughout all the visible spectrum? Do they have significant absorption at 808 nm? The same can be said for the electrospun fibers, for which the optical transparency throughout the visible and NIR region of the electromagnetic spectrum should be analyzed.
A: We performed a spectroscopic analysis of the aqueous solution of BP@SF, which exhibit a broad range of light absorption between 600 and 900 nm. Meanwhile, to more clearly know the light absorption, the image of light absorption between 775 and 855 nm was added. 808 nm NIR was included in this study since it is generally used in photothermal therapy study [39, 40].
Here are the details.
BP has a wide spectrum of light absorption qualities. It absorbs light between 400 and 900 nm. With this good photothermal property, BP is frequently utilized as a photothermal agent in the PTT of cancer. In this study, we investigated the spectrum absorption of 0.1 mg/mL, 0.2 mg/mL, and 0.4 mg/mL BP@SF in aqueous solution between 400-900 nm (Fig. 2e1), with an emphasis on light absorption between 775 and 855 nm (Fig. 2e2 and Fig. 2e3). BP@SF had a broad range of light absorption between 400 and 900 nm, with the absorption peak at from 800 to 810 nm being the most prominent (Fig. 2e) Furthermore, as the concentration of BP@SF increased, the intensity of light absorption increased continually. Near-infrared light at 808 nm is generally utilized to test the photothermal characteristics of BP [39, 40]. As a result, NIR at 808 nm was used for further testing in this study. (Line 140 to 149 in the manuscript)
Figure 8. (e) UV-Vis-NIR absorption spectra of thin-layer BP@SF nanosheets. (e1) wavelength from 600 to 900 nm. (e2) The absorbance of 0.4mg/mL BP@SF from 775 to 855 nm. (e3) The absorbance of 0.1mg/mL BP@SF and 0.2mg/mL BP@SF from 775 to 855 nm.
Reference:
[39] Arjun Prasad Tiwari. pH/NIR-Responsive Polypyrrole-Functionalized Fibrous Localized Drug-Delivery Platform for Synergistic Cancer Therapy. ACS Appl. Mater. Interfaces. 2018, 10, 24, 20256–20270.
[40] Yanjun Zheng. Photothermally Activated Electrospun Nanofiber Mats for High-Efficiency Surface-Mediated Gene Transfection. ACS Appl. Mater. Interfaces. 2020, 12, 7, 7905–7914
Point 8: In Figure 2d, the label of the y-axis should be “Temperature Increase” rather than just “temperature”
A: The image has been revised according to this suggestion (Fig. 2d in manuscript) .
Point 9: The chemical composition of BP@SF and of the electrospun fibers should be investigated with more precise tools than only EDS performed on SEM, which give semi-quantitative results. Other approaches for the quantification of the total phosphorous content of the preparations, such as ICP-AES or ICP-MS, should be employed to confirm the EDS results.
A: Since we just want to show that the P element is evenly distributed on the electrospun membrane's surface, and the EDS can prove the results. The BP@SF preparation method in this paper is based on the study reported and the main concern of this study is application of electrospun membranes with loaded BP@SF. So, ICP-AES or ICP-MS testing will be covered in further studies if applicable.
Point 10: The BP nanosheets should be characterized by TEM in order to assess crystallinity and thickness of the nanosheets.
A: The description has been revised, and SEM and TEM image have been introduced to characterize the exfoliated BP@SF, which are marked in the manuscript. Here are the details.
BP@SF sheets were prepared by ultrasound-assisted liquid exfoliation using SF as an effective exfoliator. With a unique hydrophilic-hydrophobic structure and abundant carboxyl groups, SF can facilitate the interfacial bonding with 2D nanomaterials [35, 36]. On this basis, during the exfoliation process, SF molecules stably bound to the BP crystal surface through strong hydrophobic interactions. Meanwhile, their hydrophilic regions are exposed to water to stabilize the exfoliated BP nanosheets and prevent re-agglomeration. Ultrathin BP@SF sheets were obtained after sonication of BP powders in aqueous SF solution for 2 h. The size of BP@SF was from 200 to 400 nm (PDI=0.688) (Fig. 1a). SEM observation (Fig. 1b) and EDS analysis (Fig. 1c) showed that BP@SF were well distributed. The exfoliated BP@SF exhibited lamellar appearance and the lattice spacing of the was 0.2823 nm (Fig. 1d and Fig. 1e), which was consistent with the study reported [20]. (Line 116 to127 in revised manuscript)
Reference:
[35] Huang, X., et al., Silk fibroin-assisted exfoliation and functionalization of transition metal dichalcogenide nanosheets for antibacterial wound dressings. Nanoscale, 2017. 9(44): p. 17193-17198.
[36] Grant, A.M., et al., Silk Fibroin–Substrate Interactions at Heterogeneous Nanocomposite Interfaces. Advanced Functional Materials, 2016. 26(35): p. 6380-6392.
[20] Xiao-Wei Huang et al. Water-Based Black Phosphorus Hybrid Nanosheets as a Moldable Platform for Wound Healing Applications. ACS APPLIED MATERIALS&INTERFACES[J] 2018, 10, 35495−35502.
Point 11: In Figure 3b, how many fibers were measured for making the size distribution histograms?
A: The diameter distribution of electrospun membranes was plotted after counting fifty fibres based on the SEM image of each sam
Round 2
Reviewer 1 Report
The manuscript was successfully improved. Only one comment: Fig 2d. It is not transparence... transmittance is the right term.
Reviewer 3 Report
All the issues highlighted by this referee have been addressed.
I suggest for the approval of the manuscript in the present form.